# Computational Modeling of Polymer Matrix Based Textile Composites

**DOI:** 10.3390/polym14163301

**Published:** 2022-08-13

**Authors:** Michal Šejnoha, Jan Vorel, Soňa Valentová, Blanka Tomková, Jana Novotná, Guido Marseglia

**Affiliations:** 1Department of Mechanics, Faculty of Civil Engineering, Czech Technical University in Prague, 166 29 Prague, Czech Republic; 2Department of Material Engineering, Faculty of Textile Engineering, Technical University of Liberec, 461 17 Liberec, Czech Republic; 3High Technical School of Architecture, University of Seville, 41012 Seville, Spain; 4Departamento de Matemática Aplicada I, University of Seville, 41012 Seville, Spain

**Keywords:** polymer matrix, basalt fibers, woven composite, Leonov model, Mori–Tanaka method, multiscale computational homogenization

## Abstract

A simple approach to the multiscale analysis of a plain weave reinforced composite made of basalt fabrics bonded to a high performance epoxy resin L285 Havel is presented. This requires a thorough experimental program to be performed at the level of individual constituents as well as formulation of an efficient and reliable computational scheme. The rate-dependent behavior of the polymer matrix is examined first providing sufficient data needed in the calibration step of the generalized Leonov model, which in turn is adopted in numerical simulations. Missing elastic properties of basalt fibers are derived next using nanoindentation. A series of numerical tests is carried out at the level of yarns to promote the ability of a suitably modified Mori–Tanaka micromechanical model to accurately describe the nonlinear viscoelastic response of unidirectional fibrous composites. The efficiency of the Mori–Tanaka method is then exploited in the formulation of a coupled two scale computational scheme, while at the level of textile ply the finite element computational homogenization is assumed, the two-point averaging format of the Mori–Tanaka method is applied at the level of yarn to serve as a stress updater in place of another finite element model representing the yarn microstructure as typical of FE${}^{2}$ based multiscale approach. Several numerical simulations are presented to support the proposed modeling methodology.

## 1. Introduction

Ever-increasing special demands on structural components employ researchers in material engineering in search for material systems with exceptional properties. In particular, their mechanical and physical properties must be continuously improved to compete with the related technological progress. This also includes economical as well as ecological aspects. Textile composites appear as a suitable candidate to meet these requirements in a variety of engineering areas. Plain weave textile composites, where fibers are located in two perpendicular directions, guarantee high stiffness, strength and durability [1] together with relatively simple manipulation when large scale composite products are prepared [2]. Modifying the mechanical properties of the matrix by adding inorganic fillers from recycled materials [3] may considerably improve the weight to stiffness ratio eventually leading to economically admissible designs with a sufficient bearing capacity. Exploiting green matrices would then allow us to design products which also meet the ecological requirements.

An appropriate choice of individual components opens the door for creating products with entirely unique characteristics. This calls for considerable activity in both experimental and computational research as finding an optimal material composition is not an easy task [4]. Carbon fibers in various forms of reinforcement embedded in a polymer matrix are widely spread in constructions and products in many different branches such as aviation, shipbuilding, space industry, automotive engineering, sport equipment etc. However, the final product is generally financially demanding, which impels both manufacturers and researchers to seek for other less expensive solutions. In this regard, the basalt fibers seem to offer a promising alternative in some applications owing to their acceptable mechanical and physical properties, while considerably reducing the price of a particular product.

Basalt fibers as reinforcing material of composites have recently been exploited in the production of composites. The mechanical properties of such a composite are similar to the properties of composites reinforced with S-glass fibers and appear better than those with E-glass fiber reinforcement. Experimental results presented in [5] showed basalt reinforced epoxy composites as a material with higher tensile, flexural and compressive Young’s modulus in comparison to E-glass fiber-based systems. The same has been observed also for compressive and bending strength, impact force and energy. Owing to better bonding between basalt fibers and epoxy resin and high interlaminar shear strength they seem to be a good option in many applications [5,6]. Therein, a higher stiffness in tension, flexure and compression has also been proposed for laminates. With respect to fatigue behavior, Dorigato and Pegoretti in [7] marked the basalt fiber-based laminate as a structural element with a high capability of sustaining progressive damage. The tensile strength was found again higher than that of the glass fiber-based laminate and comparable to that of laminates reinforced with carbon fibers. When discussing the properties of basalt fiber-based composites the curing temperature is worth mentioning as this may have a great impact on the composite mechanical response [8,9].

Although the strength and stiffness of the composite are merely driven by the reinforcement, the overall behavior of a structural product including its durability might significantly be influenced by the choice of the matrix phase. The most common matrix material used in the production of composites is epoxy resin for its specific properties, such as hydrophobic properties, temperature, UV, and ozone resistance, and resistance to oxidation by atmospheric oxygen. Furthermore, it is inert with respect to other materials, biologically inert, non corrosive and has good electric-insulating properties [10]. The market offers a relatively large variety of epoxy resins to serve as matrices in the production of composite materials. Among others, the L285 epoxy resin has recently been the subject of research aimed at potential improvement of its properties by adding a suitable filler [3].

In general, the mechanical response of various epoxy resins may considerably vary. Apart from common viscoelastic or nonlinear viscoelastic behavior, one may observe a transition to brittle-viscous or even brittle response with no viscose effects [11]. Obviously, such a different response not only influences the behavior and thus the range of applications of the corresponding composite material, but also drives the choice of the numerical method when we wish to make any predictions computationally. The geometrical complexity of the reinforcement as well as the inability to represent the overall behavior of the composite by the same constitutive model as adopted for individual components often requires a fully coupled homogenization. This is traditionally achieved with the help of FE${}^{2}$ approach [12]. With reference to textile composites the finite element homogenization [13] is carried out both at the level of yarns and textile ply adopting a suitable computational model on each scale, e.g., the periodic unit cell extracted from the composite on a given scale [14]. However, this approach might prove computationally demanding suggesting the application of more efficient methods. These fall within the category of classical micromechanical models such as the Self-consistent and Mori–Tanaka methods [15]. The Mori–Tanaka method [16,17] in particular enjoys considerable attention as being fully explicit and easy to implement, see, e.g., [18,19,20] for some recent applications. It also allows for a simple extension in the framework of Dvorak’s transformation field analysis [21] to incorporate various sources of nonlinearities arising once loading the composite beyond the elastic regime.

These opening paragraphs motivated the content of the paper. The current interest in basalt fibers influenced the choice of textile reinforcements. Their potential exploitation in advanced structural elements such as wind turbine blade [22] called for the application of high-performance synthetic epoxy resins. Expecting good physiological compatibility and superior mechanical properties the L285 epoxy resin has been found worthy of investigation. While experimental research of the resulting composite at the structural level is crucial, the initial designs are often based on numerical predictions. In light of this, an efficient computational model providing a reliable macroscopic response plays an important role. To this end, we propose a general methodology outlined within the following steps:Identifying and calibrating a suitable constitutive model of the matrix phase.Providing a remedy of the Mori–Tanaka (MT) micromechanical model to yield the yarn response comparable to that delivered by the finite element method (FEM). In this regard, the finite element (FE) predictions are taken as a sufficiently accurate representation of the composite response substituting the actual measurements in the framework of virtual experiments.Developing an efficient computational scheme based on a two-scale modeling strategy where the Mori–Tanaka method is adopted as a stress updater at the level of macroscopically homogeneous yarns.

The last two items are described in detail in Section 3 clearly identifying the novelty of this work seen in the reformulation of the original format of the Mori–Tanaka method and its application in the multi-scale computational framework of basalt babric reinforced composites.

## 2. Materials

The present section introduces the basic properties of individual phases including their elastic parameters, geometrical details of the basalt fabric, and computational models taken from the authors’ previous work, see, e.g., [14,23] for details.

### 2.1. Matrix Phase

As pointed out in the introductory part, the L285 epoxy resin [24] supplied by Havel Composites CZ s.r.o., Přáslavice, Czech Republic [25] was selected as a representative of a low-viscosity laminating resin. The process of lamination is characterized by a very short curing time, even at low temperatures. The optimum processing temperature ranges between 15 and 25 °C, while different temperatures and humidities do not have a significant effect on the strength of the curing product. The mixing ratio as recommended in the technical sheet should be preserved to avoid incomplete curing. Two particular hardeners H500 and H508 with the same mixing ratio by weight of 100:40 (resin:hardener) were examined. Chemical specifications of the epoxy resin and individual hardeners are taken from the company technical sheets and summarized in Table 1 for the sake of completeness, see also [11] for further details.

All tested specimens, see Figure 1, were produced in a standard way by first mixing the epoxy resin at room temperature for about 10 min with a corresponding hardener via magnetic stirrers. The mixture was then cured in a mold for about 24 h at room temperature and subsequently post-cured for additional 15 h at higher temperature of 60 °C. Chemical composition of the resulting polymer matrices thus differed only in the type of the adopted hardener. Potential differences in the material response within a given set of samples can, therefore, be attributed to geometrical variability of the tested specimens only, see Table 2.

The matrix phase is assumed isotropic and its properties were estimated from the results of the calibration process outlined in detail in Section 3. Here we only note in advance that the elastic shear modulus of the matrix phase used in all numerical simulations was found based on the adopted generalized Leonov model as
(1)$$G_{m}^{\mathrm{el}}=\sum_{\mu =1}^{N}G_{\mu },$$
where subscript *m* stands for the matrix and $G_{\mu }$ represent the shear stiffnesses of the Maxwell chain model. In accord with the generalized Leonov model the bulk modulus $K_{m}^{\mathrm{el}}$ is kept constant and follows from $G_{m}^{\mathrm{el}}$ as
(2)$$K_{m}^{\mathrm{el}}=\frac{2G_{m}^{\mathrm{el}}\left(1+\nu_{m}\right)}{3(1-2\nu_{m})},$$
assuming the initial value of Poisson’s ratio $\nu_{m}=0.34$.

### 2.2. Basalt Fibers

Figure 2a provides an illustrative example of an eight-layer basalt fabric/epoxy matrix based composite [23]. A sufficiently large representative binary image of the yarn cross-section is then plotted in Figure 2b.

The actual image analysis device used for structural image acquisition and analysis to arrive at Figure 2b consisted of NIKON ECLIPSE E 600 microscope (Nikon Instruments, Inc., New York, United States), Märzhauser motorized scanning stage, and digital monochrome camera VDC 1300C. The resulting gray-scale image seen in the left bottom corner of Figure 2b was then directly converted into its binary counterpart. However, this image suffered from a large number of flaws and was thus used to estimate the fiber volume fraction and a radius of an ideal fiber only. This information together with the knowledge of fiber centers served to create the final binary image (black and white segment in Figure 2b) suitable for further processing. This image clearly suggests an irregular distribution of fibers in the yarn cross-section. Various types of imperfections of a random type attributed merely to the manufacturing process can also be identified in Figure 2a.

A number of approaches are available in the literature to incorporate the random nature of geometrical details on both the mesoscale (level of textile ply) and microscale (level of yarn) into the formulation of a suitable computational model typically in terms of a statistically uniform periodic unit cell (SEPUC), see e.g., [23,26,27,28] to cite a few. This, however, goes beyond the present scope. Instead, we accept a considerably more simple formulation. On microscale, we rely on the findings presented in [29] promoting systems with a sufficiently large number of fibers exceeding 50% of their volume fraction to be represented by the periodic hexagonal array (PHA) model displayed in Figure 3.

On mesoscale, a single-ply periodic unit cell (PUC) of a plain weave textile composite evident in Figure 4 is considered. The geometrical details are taken from the literature [14] and the present material system is summarized in Figure 4d for the sake of completeness.

The binary image in Figure 2b and geometrical parameters in Figure 4d resulted in the fiber volume fraction in the yarn $c_{f}=0.56$ and the volume fraction of yarn within the textile ply $c_{y}=0.54$.

The elastic Young moduli *E* of the fiber phase were estimated from nanoindentation. The load-controlled quasi-static indentation was performed with the load function consisting of “loading” and “unloading” segments lasting 5 s each with an in-between 300 s segment of “holding” time. The maximum applied force of 2500 μN was used. This procedure was applied to several positions on the sample. The resulting reduced moduli, corresponding to longitudinal and transverse indentations into the fibers, were derived from the unloading part of the indentation curve. Next, the required Young moduli were calculated based on the contact mechanics of colliding solid bodies [30,31] assuming the Poisson ratio ν = 0.24 and a diamond tip with material parameters *E* = 1140 GPa and ν = 0.07. Taking into account all indents gave the average values of the Young modulus equal to 69.68 GPa and 64.82 GPa for the longitudinal and transverse fiber directions, respectively. Further details including basic statistics are available in Table 3. Unlike the matrix phase, the basalt fibers are assumed transversely isotropic as evident from Table 4. Point out that subscripts *L* and *T* stand for the longitudinal and transverse directions, respectively.

## 3. Methods

The present section summarizes the theoretical background needed for a nonlinear viscoelastic modeling of plain weave textile reinforced polymer matrix composites. We begin with the formulation and calibration of the L285 epoxy resin. Essential steps to perform two-step homogenization are outlined next.

### 3.1. Formulation and Calibration of Generalized Leonov Model

It has been experimentally observed that polymers show, in general, a negligible volumetric strain during plastic flow. This is supported in [32] where the application of the generalized Leonov model proved useful in the modeling of such materials. This is also why this constitutive model is exploited in the present work. As details of the model can be found in a number of contributions, see, e.g., [33,34], we address the theoretical grounds of the model only shortly.

The stepping stone in the formulation of the Leonov model is the Eyring flow equation representing the plastic component of the shear strain rate in the form
(3)$$\frac{\;de_{p}}{\;dt}=\frac{1}{2A}\sinh\left(\tau /\tau_{0}\right).$$

The total shear strain rate combining the elastic and plastic strain rates then becomes
(4)$$\frac{\;de}{\;dt}=\frac{\;de_{e}}{\;dt}+\frac{\;de_{p}}{\;dt}=\frac{\;de_{e}}{\;dt}+\frac{\tau }{\eta (\;de_{p}/\;dt)},$$
which is the one-dimensional Leonov constitutive model [35] with the shear-dependent viscosity η given by
(5)$$\eta \left(\;de_{p}/\;dt\right)=\frac{\eta_{0}\tau }{\tau_{0}\sin h\left(\tau /\tau_{0}\right)}=\eta_{0}a_{\sigma }\left(\tau \right),$$
where τ is the shear stress and $A,\tau_{0}$ are the model parameters, $\eta_{0}$ is the viscosity corresponding to a linear viscoelastic response and $a_{\sigma }$ is the stress-dependent shift factor. Notice that Equation (Equation 4) represents a single Maxwell unit with variable viscosity. To describe the material response sufficiently accurately, the generalized Maxwell chain model is typically used. Extension to multidimensional behavior introduces an equivalent deviatoric stress $\tau_{eq}$ as
(6)$$\tau_{eq}=\sqrt{J_{2}}=\sqrt{\frac{1}{2}s{}^{T}Q^{-1}s},\;Q=\mathrm{diag}\left[1,1,1,\frac{1}{2},\frac{1}{2},\frac{1}{2}\right],$$
where *s* is the deviatoric stress vector. The viscosity of the $\mu_{th}$ unit is then provided by
(7)$$\eta_{\mu }=\eta_{0,\mu }a_{\sigma }\left(\tau_{eq}\right)=\frac{\eta_{0,\mu }\tau_{eq}}{\tau_{0}\sin h\left(\tau_{eq}/\tau_{0}\right)}.$$

Admitting material isotropy, small strain theory, and the bulk response to be linearly elastic we arrive at the complete set of constitutive equations defining the compressible generalized Leonov model
(8)σm=Kεv,(9)dsdt=∑μ=1M2GμQdedt−dep,μdt,s=∑μ=1Msμ,(10)sμ=2ημQdep,μdt=2η0,μaσ(τeq)Qdep,μdt,
where $\sigma_{m}=\frac{1}{3}\left(\sigma_{x}+\sigma_{y}+\sigma_{z}\right)$ is the means stress, $\varepsilon_{v}=\left(\varepsilon_{x}+\varepsilon_{y}+\varepsilon_{z}\right)$ is the volumetric strain, *K* is the material bulk modulus, $G_{\mu }$ is the shear modulus, associated with the μ-th unit, $e=\varepsilon -\frac{1}{3}m\varepsilon_{v}$ stores the components of the deviatoric strain and $m{}^{T}=\left\{1,1,1,0,0,0\right\}$.

#### 3.1.1. Tensile Tests at Different Strain Rates

The most straightforward calibration of the stress shift factor $a_{\sigma }$ is to construct an Eyring plot [33,36] assuming that at plastic yielding the plastic strain rate equals the total strain rate. The yield stress is then defined as a stress level, which remains constant at further straining. With reference to the Leonov model, we recognize the analogy with an elastic perfectly plastic von Mises material. Thus, beyond the yield point, the material behaves as a generalized Newton fluid
(11)$$\sigma =m\sigma_{m}+2\eta \left({\dot{E}}_{d}\right){\dot{e}}_{p}=m\sigma_{m}+2\eta \left({\dot{E}}_{d}\right)\dot{e},$$
where the notation $\dot{e}=\frac{\;de}{\;dt}$ was introduced for the sake of conciseness and ${\dot{E}}_{d}=\sqrt{2\dot{e}{}^{T}Q\dot{e}}$ is the rate of equivalent deviatoric strain. In simple tension and at plastic yielding ($\sigma_{x}=f_{y}$) we get
(12)$${\dot{\varepsilon }}_{v}=0,\;{\dot{E}}_{d}=\sqrt{3}{\dot{\varepsilon }}_{x},\;\tau_{eq}=\frac{1}{\sqrt{3}}f_{y}.$$

With reference to Equation (Equation 3) and realizing that $2{\dot{e}}_{p}={\dot{\gamma }}_{p}$ is equivalent to ${\dot{E}}_{d}$ in plastic yielding and multidimensional space we may write
(13)$$f_{y}=\tau_{0}\sqrt{3}\mathrm{arcsinh}\left(A\sqrt{3}{\dot{\varepsilon }}_{x}\right),$$
which for large values of $A{\dot{E}}_{d}$ simplifies as
(14)$$f_{y}=\tau_{0}\sqrt{3}\ln\left(2A\sqrt{3}\right)+\tau_{0}\sqrt{3}\ln{\dot{\varepsilon }}_{x}.$$

Equation (Equation 14) thus suggests that parameters of the Eyring flow model $A,\tau_{0}$ can be found through a linear regression in the $f_{y}\times {\dot{\varepsilon }}_{x}$ diagram.

The specimens in Figure 1 were loaded in the displacement control regime at a specific strain rate until failure using the MTS Alliance 30 kN (MTS, Eden Prairie, Minnesota, United States) electromechanical testing machine equipment with 30 kN load cell. The evolution of strain $\varepsilon_{x}$ was measured using a clip on extensometer with an initial gauge length of 25 mm tightly mounted on the surface of the tested specimen, see Figure 5b. The corresponding stresses were derived by dividing the chamber force by the average cross-section area obtained from several measurements along the specimen length, recall Table 2.

#### 3.1.2. Creep Tests at Different Stress Levels

We open this section by assuming that the creep compliance function J(t) can be well approximated by the Dirichlet series as
(15)$$J\left(t\right)=\sum_{\mu =1}^{M}J_{\mu }\left[1-\exp\left(-\frac{t}{\tau_{\mu }a_{\sigma }\left(t\right)}\right)\right],$$
where $\tau_{\mu }$ are the selected retardation times. Note that the first term is typically assumed sufficiently small to represent in the limit τ→0 an elastic solid. The compliances $J_{\mu }$ of the Kelvin units with nonlinear viscosities $\eta_{\mu }\left(\tau_{eq}\right)=\tau_{\mu }J_{\mu }a_{\sigma }\left(\tau_{eq}\right)$ are derived by matching Equation (Equation 15) with the experimentally derived Master curve.

The required data are provided by standard creep tests performed at different stress levels thus exploiting the time—stress superposition principle consistent with the Eyring flow model. In the present study, the MTS Mini Bionix 858.02 testing system equipped with 1000 N load cell was used, see Figure 5b. The specimens of the same geometry as used in the tensile tests were loaded by a constant force corresponding to stresses in the range of 10 to 60 MPa. The specimen preload was carried out at a constant loading speed of 500 Ns${}^{-1}$. The strains were then recorded for two hours using again the 25 mm gauge length extensometer. To construct the creep compliance function from the above tests we adopted the following steps:(1)Transform $t\times \varepsilon_{x}$ data into $t\times e_{x}$ plots where *t* is the time in [s] and $e_{x}$ is the deviatoric normal strain provided by
(16)$$e_{x}=\varepsilon_{x}-\frac{1}{3}\varepsilon_{v}=\varepsilon_{x}-\frac{\sigma_{m}}{K},\;\sigma_{m}=\frac{1}{3}\sigma_{x},\;K=\frac{E}{3(1-2\nu )},$$
where $\varepsilon_{v}$ is the volumetric strain and $\sigma_{m}$ is the mean stress, recall Equation (8). The bulk modulus *K* is calculated from the elastic modulus *E* estimated from the elastic part of the stress–strain curve when preloading the specimen to the desired stress level. The Poisson ratio ν is assumed to be known.(2)Transform the $t\times e_{x}$ curves into $t\times \tau_{eq}$ data where $\tau_{eq}$ is provided by
(17)$$\tau_{eq}=\frac{2e_{x}}{s_{x}},\;s_{x}=\sigma_{x}-\sigma_{m}=\frac{2}{3}\sigma_{x}.$$(3a)Shift the $t\times \tau_{eq}$ data along the *t*-axis using the corresponding shift factor $a_{\sigma }\left(\tau_{eq}\right)$ obtained previously from tensile tests, recall Section 3.1.1. This is achieved by multiplying the original time by $a_{\sigma }\left(\tau_{eq}\right)$. This way we arrive at the creep compliance curve (Master curve) we would obtain for a sufficiently low stress that produces a viscoelastic response and is maintained for a long time.(3b)Alternatively, as will yet be proven useful, the Eyring flow model parameter $\tau_{0}$ and thus the shift factor $a_{\sigma }$ can be estimated directly from the creep tests. To this purpose, a simple optimization algorithm was implemented in MATLAB (vr. R2021b) software. In particular, the search for $\tau_{0}$ defining the shift factor $a_{\sigma }$ via Equation (Equation 5) amounts to finding a minimum of a single-variable function of $a_{\sigma }$ on a fixed interval. When formulating the objective function to be minimized we remember that the creep compliance functions, when shifted horizontally based on a given value of $a_{\sigma }$, partially overlap. In light of this, the objective function was defined as the square root of the sum of squares of deviations between the experimental data points, corresponding to consecutive load levels, over the current overlapping region. All curves were exploited at once. The minimization was performed with the help of the MATLAB function *fminbnd*.

As already mentioned the search for the compliances $J_{\mu }$ represents the second minimization problem written again in the framework of the least square method. To that end, we compare a certain set of experimentally measured values of the compliance function J(t) with those provided by Equation (Equation 15). To arrive at meaningful results, the first term in the Dirichlet series expansion $J_{1}$ was constrained to represent the elastic limit estimated from the initial slope of the creep test. It should be mentioned that the approximation of creep compliance function using Equation (Equation 15) is valid only for the time range specified by the retardation times $\tau_{\mu }$.

Application of Equation (9) calls for the transformation of creep compliance function (Equation 15) to relaxation function R(t) given by
(18)$$R\left(t\right)=\sum_{\mu =1}^{M}G_{\mu }\exp\left(-\frac{t}{\theta_{\mu }a_{\sigma }\left(t\right)}\right).$$

Typically, the Laplace transformation is employed to obtain the relaxation times $\theta_{\mu }$ and the shear stiffnesses $G_{\mu }$ of the Maxwell units, see, e.g., [37].

The theoretical formulation of the constitutive model of the matrix phase can be now completed by integrating Equation (9) in time. For simplicity, the most simple, those only conditionally stable, fully explicit forward Euler scheme is employed. Provided that the total strain rate is constant during the integration of a new state of stress in the matrix phase at the end of the current time step, Δt assumes the form
(19)$$\begin{array}{ccc} \sigma_{m}\left(t_{i}\right) & = & \sigma_{m}\left(t_{i-1}\right)+K\Delta \varepsilon_{v}, \end{array}$$
(20)$$\begin{array}{ccc} s(t_{i}) & = & s\left(t_{i-1}\right)+2\hat{G}\left(t_{i-1}\right)Q\Delta e+\Delta \lambda \left(t_{i-1}\right), \end{array}$$
where $t_{i}$ is the current time at the end of the *i*-th time increment. In light of Equation (Equation 18) the instantaneous shear modulus $\hat{G}$ and the increment of eigenstress Δλ read
(21)$$\begin{array}{ccc} \hat{G} & = & \sum_{\mu =1}^{M}G_{\mu }\frac{\theta_{\mu }a_{\sigma }\left(t_{i-1}\right)}{\Delta t}\left[1-\exp\left(-\frac{\Delta t}{\theta_{\mu }a_{\sigma }\left(t_{i-1}\right)}\right)\right], \end{array}$$
(22)$$\begin{array}{ccc} \Delta \lambda & = & -\sum_{\mu =1}^{M}\left[1-\exp\left(-\frac{\Delta t}{\theta_{\mu }a_{\sigma }\left(t_{i-1}\right)}\right)\right]s_{\mu }\left(t_{i-1}\right). \end{array}$$

Further details can be found, e.g., in [38].

### 3.2. Computational Homogenization

To introduce the subject we consider a representative volume element (RVE) of a single-ply balanced plain weave composite plotted in Figure 4 with a potential distribution of fibers in the transverse cross-section being represented by a periodic hexagonal array model as plotted in Figure 3. These simple geometrical representations allow for a standard periodic homogenization to be exploited on the mesoscopic (ply) as well as microscopic (yarn) scales.

When limiting attention to elasticity, a simple bottom-up fully uncoupled homogenization can be used to predict the macroscopic effective properties of a single-ply textile composite. This, however, is no longer possible when loading the composite beyond the elastic limits as the material anisotropy precludes derivation of the homogenized nonlinear constitute law at the level of yarn. A two-step, fully coupled homogenization, is therefore needed. Two options are typically available:Analysis in the framework of FE${}^{2}$ approach. In this case, the finite element homogenization is carried out both at the level of yarns and textile ply adopting a suitable computational model on each scale, e.g., the periodic unit cells in Figure 3b and Figure 4b. In this approach, the ply RVE (mesoscale) is typically loaded by the prescribed macroscopic strains or stresses, while the yarn RVE (microscale) is subjected to strains developed at an integration point of a finite element located in the yarn. Upon return, the updated stress averages and potentially the modified yarn effective properties are transferred back to the mesoscale.The FE${}^{2}$ approach might be computationally too demanding. This suggests a combination of the finite element method at the level of textile ply and the computationally efficient Mori–Tanaka method which replaces the finite element analysis at the level of yarns. This approach in particular is examined in the present study. A graphical representation of this two-step modeling strategy, where $\Delta \bar{\varepsilon }$ is the mesoscopic strain at a given point of the mesoscopic finite element mesh and $\Delta \bar{\sigma }$ is the corresponding mesoscopic stress increment provided by the Mori–Tanaka method, is shown in Figure 6. The eigenstrain Δμ might be associated with many physical sources, but only the viscoelastic effects are considered hereinafter.

Both homogenization approaches, FE and MT method based, will be now briefly reviewed. Only the most relevant formulations will be outlined. Thus, for more details, the interested reader is referred to [14,15].

#### 3.2.1. First Order Homogenization Using Finite Element Method

The presented first-order homogenization grounds on the formulation developed in [13]. To this end, suppose that a periodic unit cell *Y*, representing all the geometrical and material details of the whole composite, is loaded on its outer boundary by the prescribed displacements or tractions that produce the macroscopically uniform strains *E* or stresses Σ in an equivalent homogeneous medium. The macroscopic constitutive equations then read
(23)$$\Delta \Sigma =L^{\mathrm{hom}}\Delta E+\Delta \Lambda ,\;\Delta E=M^{\mathrm{hom}}\Delta \Sigma +\Delta \Upsilon ,$$
where $L^{\mathrm{hom}}$ and $M^{\mathrm{hom}}$ are the instantaneous effective (homogenized) stiffness and compliance matrices, respectively, and ΔΛ, and ΔΥ are the corresponding eigenstresses and eigenstrains. We choose an incremental format in view of the nonlinear viscoelastic model described by Equations (20)–(22). The macroscopic strains and stresses are related to volume strains and stress averages of local fields developed in individual phase r=f(fiber,yarn),m(matrix) as
(24)$$\Delta E=\langle \Delta \varepsilon (x)\rangle =\sum_{r}^{2}c_{r}\Delta \varepsilon_{r},\;\Delta \Sigma =\langle \Delta \sigma (x)\rangle =\sum_{r}^{2}c_{r}\Delta \sigma_{r},$$
where $\langle \cdot \rangle$ stands for volume averaging, $c_{r}$ is the volume fraction of a given phase *r* and $\Delta \varepsilon_{r}$ and $\Delta \sigma_{r}$ are increments of piece-wise uniform phase strains and stresses, respectively, see [14] for further details. Similar to Equation (Equation 23), the local fields can be written in terms of the phase material stiffness $L_{r}$ and compliance $M_{r}$ matrices as
(25)$$\Delta \sigma_{r}=L_{r}\Delta \varepsilon_{r}+\Delta \lambda_{r},\;\Delta \varepsilon_{r}=M_{r}\Delta \sigma_{r}+\Delta \mu_{r},$$
where $\Delta \mu_{r}$ and $\Delta \lambda_{r}=-L_{r}\Delta \mu_{r}$ are increments of local phase eigenstrains and eigenstresses. In the present study, only the viscoelastic strain $\Delta \mu_{m}$ developed in the matrix will be considered.

Equation (Equation 23) follows directly from Hill’s lemma [39]. Hill proved that for the assumed uniform loading conditions the volume average of the internal work (virtual work) done by local fields equals the internal work (virtual work) done by their macroscopic counterparts. This statement is mathematically written as
(26)$$\langle \delta \varepsilon \left(x\right){}^{T}\Delta \sigma \left(x\right)\rangle =\delta E{}^{T}\Delta \Sigma .$$

Because in the displacement-based finite element method the primary unknowns are the nodal displacements, we continue by decomposing the local displacements, while taking into account the periodicity of the unit cell, into a homogeneous ΔE·x and fluctuations $\Delta u^{*}$ parts as
(27)$$\Delta u\left(x\right)=\Delta E\cdot x+\Delta u^{*}\left(x\right),\;\Delta \varepsilon \left(x\right)=\Delta E+\Delta \varepsilon^{*}\left(x\right),$$
where Equation (Equation 27)${}_{2}$ follows from standard strain-displacement relations. Next, substituting from Equation (Equation 27)${}_{2}$ into Equation (Equation 26) and realizing that
(28)Δσ(x)=L(x)Δε(x)+Δλ(x),
provides Hill’s lemma in the form
(29)$$\delta E{}^{T}\langle L\left(x\right)(\Delta E+\Delta \varepsilon^{*}\left(x\right))+\Delta \lambda \left(x\right)\rangle +\langle \delta \varepsilon^{*}{}^{T}\left[L\left(x\right)\left(\Delta E+\Delta \varepsilon^{*}\left(x\right)\right)+\Delta \lambda \left(x\right)\right]\rangle =\delta E\Delta \Sigma .$$

Because the variations δE and $\delta \varepsilon^{*}$ are independent, Equation (Equation 29) can be split into two equalities
(30)$$\begin{array}{ccc} \delta E{}^{T}\Delta \Sigma & = & \delta E{}^{T}\left(\langle L(x)\rangle \Delta E+\langle L\left(x\right)\Delta \varepsilon^{*}\left(x\right)\rangle +\langle \Delta \lambda (x)\rangle \right), \end{array}$$
(31)$$\begin{array}{ccc} 0 & = & \langle \delta \varepsilon^{*}{}^{T}L\left(x\right)\rangle \Delta E+\langle \delta \varepsilon^{*}{}^{T}L\left(x\right)\Delta \varepsilon^{*}\left(x\right)\rangle +\langle \delta \varepsilon^{*}{}^{T}\Delta \lambda \left(x\right)\rangle . \end{array}$$

Equations (Equation 30) and (31) are to be solved for unknown increments of macroscopic ΔE and fluctuation $\Delta \varepsilon^{*}$ strain fields. When deriving this set of equations we explicitly assumed that the macroscopic stress increment ΔΣ is prescribed thus considering the stress-based approach [14]. However, the strain based formulation is often needed. In such a case, the unit cell is loaded by the prescribed macroscopic strain ΔE. The virtual change of the prescribed quantity then vanishes (δE=0) and Equation (Equation 29) simplifies to
(32)$$\langle \delta \varepsilon^{*}{}^{T}L\left(x\right)\Delta \varepsilon^{*}\left(x\right)\rangle =-\langle \delta \varepsilon^{*}{}^{T}L\left(x\right)\rangle \Delta E-\langle \delta \varepsilon^{*}{}^{T}\Delta \lambda \left(x\right)\rangle .$$

In the framework of FEM the fluctuation displacements $u^{*}$ rather than the total displacements *u* will be considered as the primary unknowns. Standard finite element discretization then reads
(33)$$\Delta u^{*}\left(x\right)=N\left(x\right)\Delta r,\;\Delta \varepsilon (x)=\Delta E+B\left(x\right)\Delta r,$$
where matrix N stores the shape functions for a given partition of the unit cell, B is the corresponding geometrical matrix and *r* is the vector of unknown nodal degrees of freedom. Introducing the fluctuation strains from Equation (Equation 33) into Equation (Equation 32) provides the final set of discretized equations of equilibrium in the form
(34)$$\int_{\Omega }B\left(x\right){}^{T}L\left(x\right)B\left(x\right)\;d\Omega \Delta r=-\int_{\Omega }B\left(x\right){}^{T}L\left(x\right)\Delta E\;d\Omega -\int_{\Omega }B\left(x\right){}^{T}\Delta \lambda \left(x\right)\;d\Omega ,$$
where Ω represents the volume of the unit cell *Y*. It is worth mentioning that the fluctuation displacements $u^{*}$ must satisfy certain conditions so that the following relations hold
(35)$$\langle \Delta \varepsilon (x)\rangle =\Delta E,\;\langle \varepsilon^{*}\left(x\right)\rangle =0.$$

Point out that Equation (Equation 35)${}_{2}$ is for example satisfied when enforcing homogeneous displacements *r* on the entire boundary of the unit cell. Usually, better predictions are obtained when considering periodic boundary conditions which for a rectangular PUC amounts to enforcing the same fluctuation displacements on opposite faces of PUC. To avoid writing any master–slave type of constraints it is preferable to consider periodic meshes. The periodicity condition is then enforced simply by assigning the same code numbers to the corresponding degrees of freedom on opposite faces of PUC.

#### 3.2.2. Homogenization Based on Mori–Tanaka Method and Transformation Field Analysis

The Mori–Tanaka method belongs to the class of micromechanical models that ground on the Eshelby solution of an ellipsoidal inclusion problem where a single inclusion is imagined in an unbounded homogeneous body loaded at infinity by a uniform stress or strain [40] fields. Eshelby showed that in such a case the distribution of inclusion strains and stresses is also uniform and derived a localization tensor that identifies the inclusion strain and stresses in terms of the applied far fields, material properties of the two phases and geometry of the inclusion. A number of existing micromechanical models take advantage of this result and attempt to extend the Eshelby solution to a composite with a large number of interacting inclusions. In his reformulation of the original Mori–Tanaka method [16], Benveniste [17] showed that the MT method accounts for the interaction of inclusions by introducing a single inclusion into an infinite matrix, but unlike the Eshelby solution this system is loaded by yet unknown average strains or stresses found in the matrix phase. This method is therefore explicit and for that reason enjoys popularity. As there is a voluminous literature on this subject, we do not attempt to present all the details of the method but provide just the relations needed in the context of this paper. To become more familiar with this method we point the interested reader to the following two monographs [14,15].

To be consistent with the previous section we consider again a two-phase fiber-matrix composite and write the local constitutive Equation (Equation 25)${}_{1}$ assuming elastic response of the fiber phase and viscoelastic response of the matrix as
(36)$$\Delta \sigma_{f}=L_{f}\Delta \varepsilon_{f},\;\Delta \sigma_{m}={\hat{L}}_{m}\Delta \varepsilon_{m}+\Delta \lambda_{m},$$
where $L_{r}$ (r=f,m, subscripts f,m stand for the fiber and matrix phase) is the phase material stiffness matrix and $\left(\;\hat{\cdot }\;\right)$ represents the dependence on the current viscoelastic shear modulus, recall Equation (Equation 21). The local strains in individual phases follow from the application of Dvorak’s transformation field analysis [21] and are provided by
(37)$$\Delta \varepsilon_{f}={\hat{A}}_{f}\Delta E+{\hat{D}}_{fm}\Delta \mu_{m},\;\Delta \varepsilon_{m}={\hat{A}}_{m}\Delta E+{\hat{D}}_{mm}\Delta \mu_{m},$$
where ${\hat{A}}_{r}$ and ${\hat{D}}_{rm}$ are the mechanical strain localization factors and strain and stress transformation influence functions, respectively. It can be shown [15,21] that for a two-phase composite the transformation influence functions are readily provided in terms of the localization factors as
(38)$${\hat{D}}_{fm}=\left(I-{\hat{A}}_{f}\right){\left({\hat{L}}_{m}-L_{f}\right)}^{-1}{\hat{L}}_{m},\;{\hat{D}}_{mm}=\left(I-{\hat{A}}_{m}\right){\left({\hat{L}}_{m}-L_{f}\right)}^{-1}{\hat{L}}_{m}.$$

It remains to determine the mechanical strain localization factors $A_{r}$. To do so, we consider, in light of the MT method, a single inclusion of an elliptical shape being embedded into an unbounded matrix, which is loaded at infinity by the average strain in the matrix phase. In the absence of viscoelastic contribution, the strain increment in the fiber phase can be then written with the help of partial strain localization factor ${\hat{T}}_{f}$ as
(39)$$\Delta \varepsilon_{f}={\hat{T}}_{f}\Delta \varepsilon_{m}.$$

With reference to Equation (Equation 24)${}_{1}$ it is now possible to express the average matrix strain increment $\Delta \varepsilon_{m}$ in the form
(40)$$\Delta \varepsilon_{m}={\left[c_{m}I+c_{f}{\hat{T}}_{f}\right]}^{-1}\Delta E={\hat{A}}_{m}\Delta E,$$
where I is the identity matrix. Substituting the right hand side of Equation (Equation 40) back to Equation (Equation 39) finally gives
(41)$$\Delta \varepsilon_{f}={\hat{T}}_{f}{\left[c_{m}I+c_{f}{\hat{T}}_{f}\right]}^{-1}\Delta E={\hat{A}}_{f}\Delta E.$$

Without derivation, see for example [40] for details, we provide the particular form of ${\hat{T}}_{f}$, the result of the Eshelby transformation inclusion problem, as
(42)$${\hat{T}}_{f}={\left[I-\hat{P}\left({\hat{L}}_{m}-L_{f}\right)\right]}^{-1}.$$

The matrix P depends on the properties of the matrix phase and geometry of the inclusion and for the case of infinite longitudinal fibers, the only type of inclusion considered in this study, is given by [41]
(43)$$P=\left[\begin{array}{cccccc} 0 & 0 & 0 & 0 & 0 & 0\;\\ 0 & \frac{k+4m}{8m\left(k+m\right)} & \frac{-k}{8m(k+m)} & 0 & 0 & 0\;\\ 0 & \frac{-k}{8m(k+m)} & \frac{k+4m}{8m(k+m)} & 0 & 0 & 0\;\\ 0 & 0 & 0 & \frac{k+2m}{2m(k+m)} & 0 & 0\;\\ 0 & 0 & 0 & 0 & \frac{1}{2p} & 0\;\\ 0 & 0 & 0 & 0 & 0 & \frac{1}{2p} \end{array}\right]$$
where k,m,l,n,p are the Hill moduli of the matrix phase and can be written in terms of Young’s moduli and Poisson’s ratios of transversely isotropic material, remember the material parameters of the basalt fiber listed in Table 4, in the form
$$\begin{array}{ccc} k & = & -{\left[1/G_{T}-4/E_{T}+4\nu_{A}^{2}/E_{A}\right]}^{-1},\;m\;=\;G_{T},\\ l & = & 2k\nu_{A},\;n\;=\;E_{A}+4k\nu_{A}^{2}=E_{A}+l^{2}/k,\;p\;=\;G_{A}. \end{array}$$

The homogenized stiffness matrix, the overall stress and eigenstress vectors associated with a lower scale, here being represented by the level of yarn (microscale), are finally provided by substituting the local stress increments from Equation (Equation 36) into Equation (Equation 24)${}_{2}$ and using relations (Equation 37) together with $\Delta \lambda_{m}=-{\hat{L}}_{m}\Delta \mu_{m}$ to get
(44)$$\begin{array}{ccc} \Delta \Sigma & = & c_{f}\Delta \sigma_{f}+c_{m}\Delta \sigma_{m}\;=\;c_{f}L_{f}\Delta \varepsilon_{f}+c_{m}{\hat{L}}_{m}\left(\Delta \varepsilon_{m}-\Delta \mu_{m}\right)\\ & = & c_{f}L_{f}\left({\hat{A}}_{f}\Delta E+{\hat{D}}_{fm}\Delta \mu_{m}\right)+c_{m}{\hat{L}}_{m}\left({\hat{A}}_{m}\Delta E+{\hat{D}}_{mm}\Delta \mu_{m}\right)-c_{m}{\hat{L}}_{m}\Delta \mu_{m}\\ & = & \left(c_{f}L_{f}{\hat{A}}_{f}+c_{m}{\hat{L}}_{m}{\hat{A}}_{m}\right)\Delta E+\left(c_{f}L_{f}{\hat{D}}_{fm}+c_{f}{\hat{L}}_{m}{\hat{D}}_{mm}-c_{m}{\hat{L}}_{m}\right)\Delta \mu_{m}. \end{array}$$

Comparing Equation (Equation 44) with Equation (Equation 23)${}_{1}$ finally gives the homogenized stiffness matrix $L_{\mathrm{MT}}^{\mathrm{hom}}$ and the increment of mesoscopic eigenstress $\Delta \Lambda_{\mathrm{MT}}$
(45)$$\begin{array}{ccc} L_{\mathrm{MT}}^{\mathrm{hom}} & = & c_{f}L_{f}{\hat{A}}_{f}+c_{m}{\hat{L}}_{m}{\hat{A}}_{m}, \end{array}$$
(46)$$\begin{array}{ccc} \Delta \Lambda_{\mathrm{MT}} & = & \left(c_{f}L_{f}{\hat{D}}_{fm}+c_{f}{\hat{L}}_{m}{\hat{D}}_{mm}-c_{m}{\hat{L}}_{m}\right)\Delta \mu_{m}. \end{array}$$

#### 3.2.3. Modification of Original Format of Mori–Tanaka Method

The literature offers solid evidence regarding the inability of the original format of the Mori–Tanaka method (piece-wise uniform distribution of strains and stresses in individual phases) to provide results comparable to detailed numerical simulations using FEM for loading conditions exceeding the elastic limit, see [42,43,44] (for illustration).

A much stiffer response predicted by the MT method is typically observed. This can be attributed to the way the stresses are localized into the fiber phase as the matrix response is assumed to be controlled solely by the constitutive law. In FEM simulations the stress transfer between individual phases is most probably affected by the formation of shear bands in the matrix. Here, we attempt to address this issue by suitably modifying the local constitutive model of the fiber phase. Being inspired by [45,46] we suggest to reduce the stresses taken by the fibers through a damage-like parameter ω and write the fiber stress increment as
(47)Δσ˜f=LfΔεf,(48)Δsf,ij=(1−ω)(Δσ˜f,ij−δijΔσ˜f,m),Δσ˜f,m=13Δσ˜f,ii,(49)Δσf,ij=Δsf,ij+δijΔσ˜f,m,
where $\Delta s_{f},\Delta {\tilde{\sigma }}_{f,m}$ represent the deviatoric and mean components of the fiber stress increment $\Delta \sigma_{f}$, respectively. The tensorial notation in Equations (48) and (49) is adopted just for the sake of convenience. Point out that stress reduction applies to the deviatoric stress components only.

The evolution of damage parameter ω is assumed similar to that proposed in [46] but replaces the equivalent stress $\tau_{eq}$ with the viscoelastic equivalent deviatoric strain in the matrix $E_{d,m}^{ve}=\sqrt{2e_{m}^{ve}Qe_{m}^{ve}}$. Considering only the viscoelastic part of the total strain allows us to write
(50)$$\begin{array}{c} \omega =N{\left\{1-\left[\frac{E_{d,m}^{ve,t}}{E_{d,m}^{0}}/\sinh\left(\frac{E_{d,m}^{ve,t}}{E_{d,m}^{0}}\right)\right]\right\}}^{M}, \end{array}$$
where M,N are the model parameters. These parameters are found by matching the FEM predictions of the homogenized response by comparing the PHA model predictions and the MT two-phase estimates given by Equation (Equation 44). Note that the stiffness matrix of the fiber phase $L_{f}$ must be suitably modified by introducing a reduced shear modulus
(51)$${\hat{G}}_{f}=\left(1-\omega \right)G_{f}.$$

The scaling parameter $E_{d,m}^{0}$ is written in terms of $\tau_{0}$ and the elastic shear modulus $G_{m}^{el}$, recall Equation (Equation 1), as
(52)$$E_{d,m}^{0}=\frac{\tau_{0}}{G_{m}^{el}}.$$

## 4. Results

Following the general structure of the previous section, we begin with the results concerning the calibration process of the matrix phase. The results of numerical simulations starting from the level of yarns and followed by the level of textile ply are summarized next.

### 4.1. Parameters of Leonov Model from Tensile Tests

At the initial stage of our experimental program the attention was accorded to the L285 epoxy resin cured with the H508 hardener. Six specimens in particular were examined. Figure 7a shows the resulting stress–strain curves confirming a strong strain rate dependency and nonlinear viscous behavior over the whole range of the applied strain rates, which is manifested by variation of both the initial shear stiffness and yield stress $f_{y}$. The latter quantity is defined as the maximum stress attained in the experiment. The corresponding Eyring plot is presented as a dashed line in Figure 7d resulting in $\tau_{0}=1.7$ MPa.

### 4.2. Parameters of Maxwell Chain Model from Creep Tests

Expecting the lack of H508 hardener on the Czech market we performed this particular set of experiments using the same L285 epoxy resin but cured with H500 hardener as a recommended substitute. Fourteen specimens were examined. Each test, except for tests at 40 and 45 MPa, was run twice. As seen in Figure 8a, the results at low stress levels (10–30 MPa) are comparable thus supporting the measurement credibility. Negligible deviations are attributed to the evaluation procedure as an average cross-sectional area calculated from all tested samples, recall Table 2, was used to convert the measured force-displacement curve into the correspondent stress–strain curve. The maximum stress level was suggested based on the maximum yield stress observed in the tensile tests. It is evident that particularly at high stress levels the acquired measurements were greatly affected by the quality of the specimen as well as the type of hardener, compare the curves for 50 and 60 MPa. The latter influence will yet be discussed. In addition, for stress levels exceeding 40 MPa the tertiary creep can easily be identified. Exploiting these measurements would thus require a large strain formulation. Because of that, only the measurements up to 40 MPa were adopted in further processing.

When deriving the Master curve the procedure described in Section 3.1 was persuaded. However, we were not able to employ the step (3a) to simply shift the measured curves horizontally using $a_{\sigma }$ calculated on the basis of $\tau_{0}=1.7$ MPa found originally from tensile tests on L285/H508 material. Therefore, the step (3b) was eventually adopted and a significantly different value of $\tau_{0}=5.3$ MPa pertinent to L285/500 material was derived directly from creep tests. The resulting approximation of the Mater curve is plotted in Figure 8b. It is clearly seen that at t→0 the curve approaches the elastic asymptote as required.

To partially reconcile the discrepancy in the material response of the two material systems we carried out an additional set of measurements of the L285/H500 system in tension at the same strain–strain rate as originally used for the L285/H508 matrix. These results are shown in Figure 7b. An essentially brittle response at higher strain rates is evident. The Eyring plot constructed from two points only appears as a solid line in Figure 7d and with $\tau_{0}=7.3$ MPa significantly deviates from that of L285/H508 material. This is also why the Eyring flow parameter $\tau_{0}$ obtained from creep tests was adopted in all the remaining calculations thus also in the derivation of both the creep compliance function J(t), Equation (Equation 15), and the relaxation function R(t), Equation (Equation 18). For nine Kelvin units of the Maxwell chain model the values of the compliances $J_{\mu }$ are listed for the selected retardation times $\tau_{\mu }$ in Table 5 together with the corresponding shear stiffnesses $G_{\mu }$ and relaxation times $\theta_{\mu }$. It is worth mentioning that the elastic modulus of the L285/H500 matrix provided by Equation (Equation 1) closely matches the one estimated from the initial slopes of the creep tests.

To support the applicability of the adopted generalized Leonov model we performed the creep tests numerically. The results are available in Figure 9a. The dashed lines correspond to stress levels not used in the experimental part. Figure 9a shows the numerically derived Master curve. A relatively good match with experimental data is evident.

For illustration, we also reproduced the tensile test numerically. The analysis was performed in the plane stress regime using 4-node quadrilateral elements. Similarly to experimental measurements, the analysis was carried out in the displacement control regime. The geometry of the computational model was selected such as to represent the part of the specimen covered by the extensometer. The resulting finite element meshes together with boundary and loading conditions is available in Figure 10a. The comparison of numerical predictions (dashed lines) with experimental measurements (solid lines) is plotted in Figure 10b. A reasonable agreement is observed. A higher value of $\tau_{0}\left(H500-\mathrm{tension}\right)=7.3$ MPa (tensile tests) in comparison to $\tau_{0}\left(H500-\mathrm{creep}\right)=5.3$ MPa (creep tests) used in simulations is reflected in a higher value of the yield stress $f_{y}$ obtained experimentally. For completeness, the Eyring plots were constructed from tensile tests ($\tau_{0}\left(\mathrm{tension}\right)$) and calculated numerically ($\tau_{0}\left(\mathrm{creep}\right)$) are compared in Figure 10.

### 4.3. Composite Response at Level of Yarn

The first set of experiments aims at verifying the implementation of the generalized Leonov model and the Mori–Tanaka method in our in-house finite element code. Figure 11a provides the results of finite element simulations of the L285 matrix phase subjected to variable macroscopic stress rates. Although a homogeneous material is assumed the analysis is carried within the framework of first-order homogenization method in the stress control regime using Equations (Equation 30) and (31). Both in-plane tension and transverse shear, recall Figure 3, show a significant rate dependency predicted by the Leonov model. Unlike in Figure 10b, the response in tension is presented here in terms of deviatoric stress and strain $e_{x}\times s_{x}$ components. It appears considerably stiffer in comparison to the pure shear loading scenario. Given also the much higher stiffness of fibers in tension compared to the polymer matrix, we limited attention in all the remaining simulations at the level of yarns to pure transverse shear so to better address the nonlinear behavior of the matrix phase.

Considering the strain control regime, Equation (Equation 32), we first compare in Figure 11b the response predicted by both the FE and MT method for two different shear strain rates. As seen, for purely isotropic material the predictions are identical. This result together with negligible computational demands of the MT method seems appealing. However, the stress–strain diagrams plotted in Figure 12 make the general applicability of the MT method less certain. We notice a relatively strong mismatch between the macroscopic response ($\Sigma_{xy}\times E_{xy}$ diagram) provided by the FE and MT methods, while this is not so significant for the matrix phase owing to the elastic-perfectly plastic character of the Leonov model with the rate-dependent yield stress, an inadequate transition of the macroscopic stress into the fiber phase is clearly seen regardless of the applied strain rate.

A remedy is offered in Section 3.2.3 through Equations (47)–(Equation 50). To adopt this new formulation, we first fitted the model parameters M,N by matching the macroscopic response derived by FEM and the MT method for the selected shear strain rate ${\dot{E}}_{xy}=10^{-3}$s${}^{-1}$. The composite was then subjected to another two different strain rates while keeping the values of parameters M,N unaltered. The result is shown in Figure 13a suggesting the ability of the present formulation to successfully address the strain rate effect with a unique set of parameters, which would not be possible with a similar formulation presented in [46] on the basis of the current equivalent stress instead of strain. Partial explanation is provided in Figure 13b showing the delay in damage evolution, which arises naturally with the evolution of equivalent viscoelastic strain.

The evolution of local phase stresses is evident in Figure 14a. As expected, the distribution of stress $\sigma_{yx}^{m}$ in the matrix phase remains the same as the new formulation has an influence on the fiber and overall stresses only. On the other hand, the prediction of fiber stress $\sigma_{xy}^{f}$ has considerably been improved, recall Figure 12b. To further support the new formulation we examined the effect of a variable strain rate on relaxation. It is seen in Figure 14b that not only the loading branch but also the relaxation phase is captured satisfactorily.

It is fair to mention that here we accepted the prediction provided by FEM as sufficiently accurate to play the role of a virtual experiment, therefore replacing the actual laboratory measurements. These, however, are still needed to validate both computational strategies. However, this goes beyond the present scope. Furthermore, as the model is generally isotropic it remains to check whether a single set of parameters will still be sufficient to account for other, more general, loading scenarios. In this regard, a recently proposed modeling strategy, see [44], might prove to be more robust.

### 4.4. Composite Response at Level of Textile Ply

A suitably tuned new version of the MT method now opens the door to the desired multiscale modeling of a single plain weave textile ply. Such an approach should prove efficient making possible to perform an extensive parametric study for a variety of loading conditions and types of reinforcements. This would certainly be appreciated particularly at the initial stage of design of a given structural element. Nevertheless, the validity of such simulations should be checked. At this point, however, no experimental measurements performed directly on a textile ply under study are available. Therefore, we propose two simple loading scenarios enabling us to evaluate the performance of the basalt fabric reinforced polymer matrix at least qualitatively.

In particular, we compare the macroscopic response of the composite to in-plane tension and in-plane and out-of-plane shear by loading the unit cell in Figure 4 in turn by the macroscopic in-plane tensile ${\dot{E}}_{XX}$, in-plane shear ${\dot{E}}_{XY}$, and out-of-plane shear ${\dot{E}}_{XZ}$ strain rates. Two specific values of $10^{-2}$ and $10^{-4}$$s^{-1}$ are examined. Because of the adopted fully explicit forward Euler integration scheme, each value requires a different size of the minimum integration time step. In general, the higher the strain rate the smaller the time step to avoid oscillations of the computed stresses. In the present study Δt=0.01 s for $10^{-2}$$s^{-1}$ and Δt=1 s for $10^{-4}$ $s^{-1}$ strain rates are considered. We remind different time steps already used to generate the results in Figure 12.

Henceforth, the macroscopic stress–strain curve, the response at the level of textile ply, is denoted as Σ×E, while $\sigma_{m}$ denotes the average stress in the matrix and $\sigma_{y}$ the average stress in the yarn. Notice that $\Delta \sigma_{y}=\Delta \bar{\sigma }$ in Figure 6 and is provided by the MT method for a given increment of the yarn strain $\Delta \varepsilon_{y}=\Delta \bar{\varepsilon }$. Similarly, the instantaneous yarn effective stiffness matrix ${\hat{L}}_{y}^{\mathrm{hom}}$ corresponds to $L^{\mathrm{MT}}$ in Figure 6. Clearly, each element of the yarn is treated as a two-phase unidirectional fibrous composite loaded by the increment of a mesoscopic strain $\Delta \varepsilon_{y}$ computed in turn for a given increment of the macroscopic strain ΔE (strain control regime) or stress ΔΣ (stress control regime).

With the above specification, we first address the tensile loading along the *X* direction. Point out that this is the case of a unidirectional strain where $\Delta E_{XX}\neq 0$ while all other strain components vanish. Such a constrained problem inevitably generates transverse stresses as seen in Figure 15a. In this direction the response is driven by the stiffness of the fiber phase where the matrix contribution is minor. This is also why the response is more or less elastic. Figure 15a also confirms a perfect symmetry in the weft and warp directions of the selected balanced plain weave composite and transverse isotropy of the yarns. To confirm the expected elastic bulk response of the composite caused by the property of the generalized Leonov model, we plot in Figure 15b the evolution of the macroscopic mean stress $\Sigma_{m}=\frac{1}{3}\left(\Sigma_{XX}+\Sigma_{YY}+\Sigma_{ZZ}\right)$ and the macroscopic equivalent stress $\tau_{eq}=\sqrt{\frac{1}{2}S_{ij}S_{ij}},\;S_{ij}=\Sigma_{ij}-\delta_{ij}\Sigma_{m}$.

The piece-wise uniform evolution of these quantities in individual phases is displayed in Figure 16. Clearly, in accordance with the Leonov model, the volumetric response of the matrix phase is fully elastic. A slight deviation of the macroscopic volumetric stress from linearity is caused by the nonlinearity evolving in the homogenized yarn. We also notice a relatively mild effect of the applied strain rate even with the difference chosen on purpose by two orders of magnitude.

As expected, the effect of the applied strain rate is more pronounced for shear loading conditions which is evident in Figure 17. A slightly stiffer response in in-plane shear is in accord with the homogenized elastic moduli summarized for the sake of convenience in Table 6. These were extracted from the homogenized stiffness matrix provided by strain-based homogenization, Equations (Equation 32) and (Equation 34). The listed values confirm the macroscopic orthotropy of the balanced plane weave composite, which makes no difference between the warp and weft directions.

## 5. Conclusions

Since gaining notable popularity, the basalt fabric/polymer matrix based composites should be thoroughly studied both experimentally and computationally, for only then new improved designs would be broadly accepted in structural applications. The present study partially contributes to this subject by offering an efficient fully coupled two-scale computational procedure to address both the geometrical complexity of basalt reinforcement and nonlinear behavior of the matrix.

The proposed approach advocates the use of the Mori–Tanaka micromechanical model to substitute a computationally expensive finite element method when estimating local stresses and strains on the yarn scale. The fact that only the overall macroscopic response is usually of an engineering interest supports this approach as the MT method generally obviates the local details by considering the piece-wise uniform stress and strain averages only. However, even such a simplification should be reliable and generally comparable to more accurate FE predictions. This is why we introduced a certain reformulation of the original format of the MT method to closely match the mesoscopic response generated by FEM. In this context, the presented results seem promising so to accept the MT method for the stress update on the microscale (unidirectional fibrous composite representing yarn) within the multiscale analysis. In this regard, some specific features of the macroscopic response of a textile ply with reference to the adopted local constitutive laws have successfully been illustrated.

While this study is only computational and should be experimentally validated, the analysis concerning the material behavior of individual phases, the basalt fibers and L285 polymer matrix, integrated both experimental and computational components of research. A complex experimental program was executed to provide data for the calibration of the generalized Leonov model chosen to represent the nonlinear viscoelastic response of the matrix phase. Within this step, a calibration procedure for tuning the model parameters of the Eyring flow was developed solely on the basis of creep tests. The calibrated model was validated computationally by reproducing the laboratory measurements numerically. The reported results highlight the importance of properly addressing the whole system resin/hardener as the results derived for one particular hardener can hardly be transplanted to another one albeit using the same epoxy resin. 

## Figures and Tables

**Figure 1 polymers-14-03301-f001:**
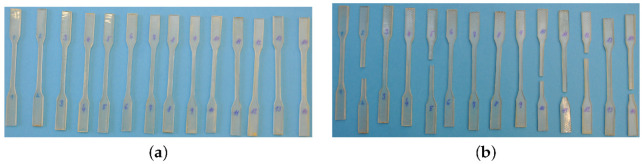
Specimens used in tensile and creep tests: (**a**) prior to testing, (**b**) after testing.

**Figure 2 polymers-14-03301-f002:**
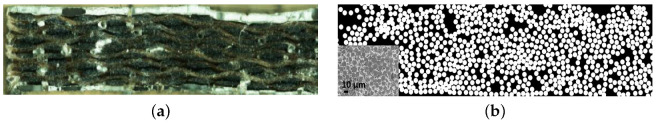
(**a**) Example of laminate made of basalt fabric bonded to epoxy matrix, (**b**) example of yarn cross-section.

**Figure 3 polymers-14-03301-f003:**
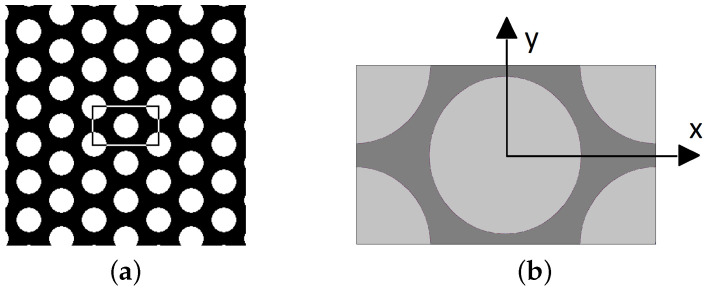
(**a**) Hexagonal arrangement of unidirectional fibers in yarn transverse direction (computationally generated image), (**b**) PHA model.

**Figure 4 polymers-14-03301-f004:**
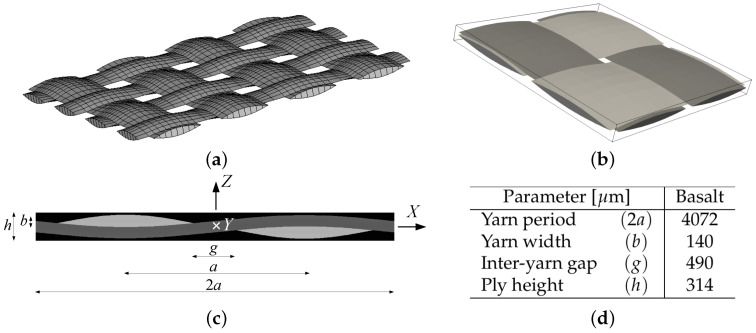
Example of periodic unit cell of a single ply textile composite: (**a**) plain weave arrangement of yarns, (**b**) plain weave PUC, (**c**,**d**) basic geometrical data.

**Figure 5 polymers-14-03301-f005:**
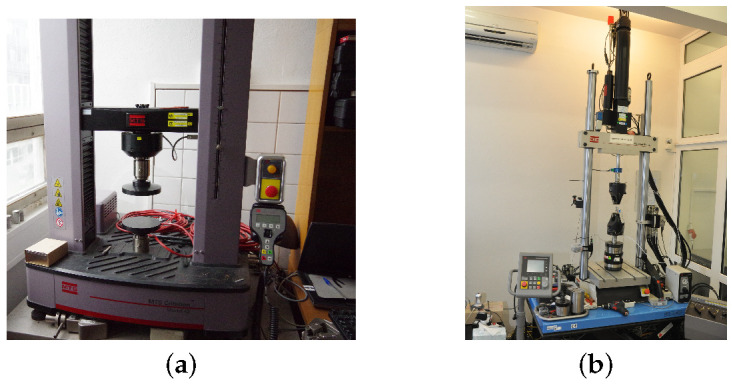
(**a**) MTS Alliance 30 kN electromechanical testing system, (**b**) MTS Mini Bionix 858.02 testing system.

**Figure 6 polymers-14-03301-f006:**
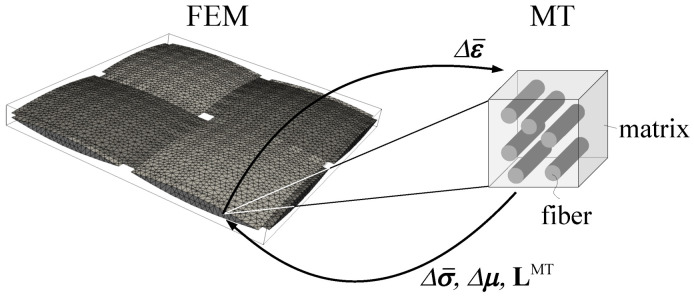
Graphical representation of two-scale computational scheme combining FEM at the level of textile ply and the MT method at the level of yarn.

**Figure 7 polymers-14-03301-f007:**
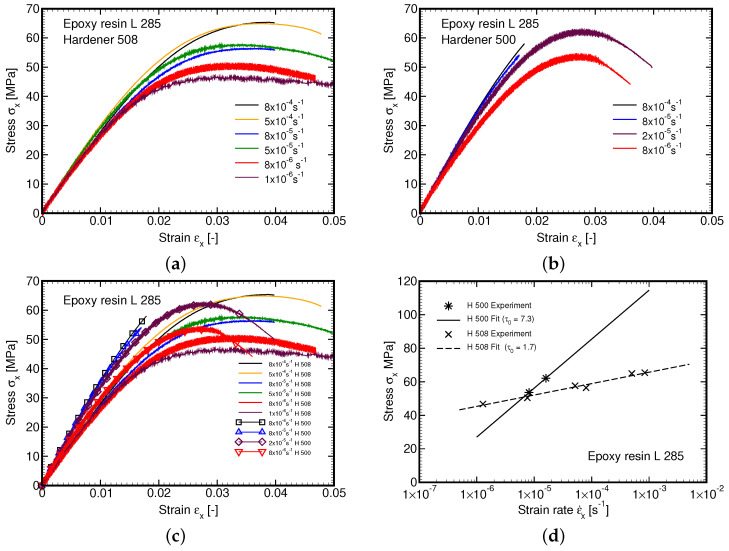
Tensile tests: (**a**) Hardener 508, (**b**) Hardener 500, (**c**) All tests, (**d**) Eyring plot.

**Figure 8 polymers-14-03301-f008:**
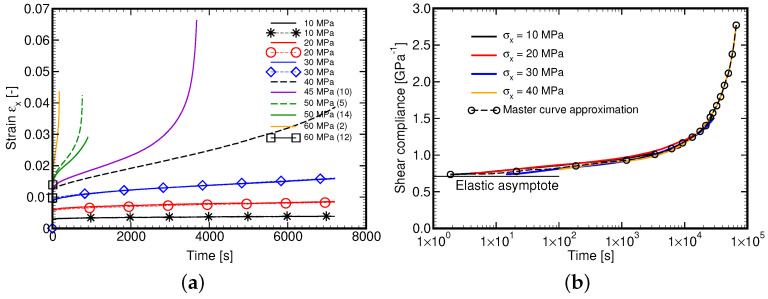
L285/H500: (**a**) Creep experiment, (**b**) Master curve derived from experiments.

**Figure 9 polymers-14-03301-f009:**
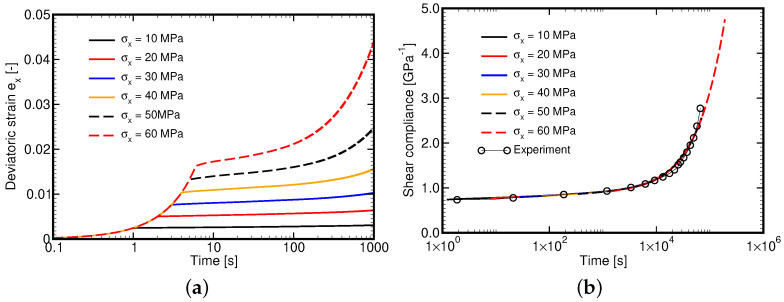
(**a**) Simulation of creep experiment, (**b**) Master curve derived from simulations.

**Figure 10 polymers-14-03301-f010:**
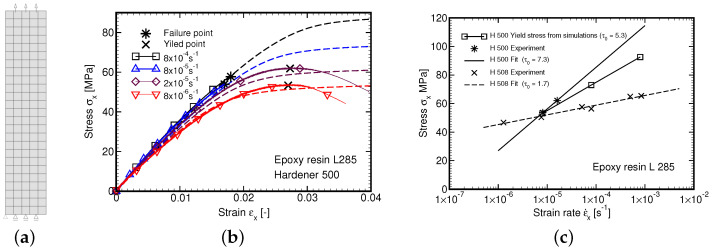
(**a**) Finite element mesh, (**b**) Simulation of tensile experiment, (**c**) Eyring plot.

**Figure 11 polymers-14-03301-f011:**
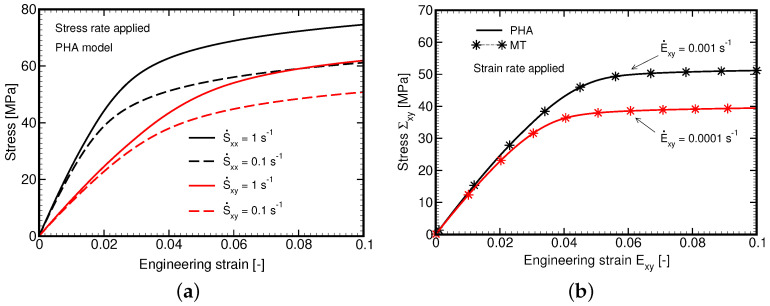
Nonlinear viscoelastic response of L285 (**a**) Stress–strain diagrams for two different strain rates provided by FEM (stress control loading); (**b**) Stress–strain diagrams for two different strain rates provided by FE and MT methods (strain control loading).

**Figure 12 polymers-14-03301-f012:**
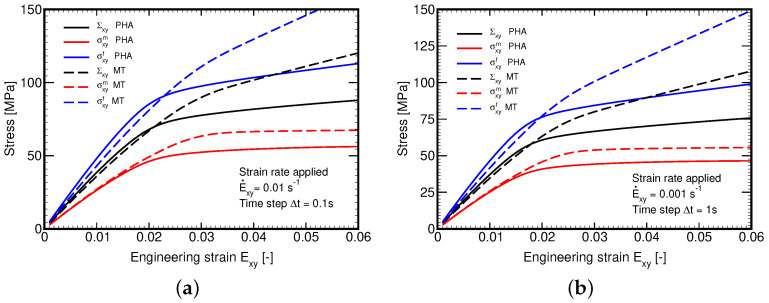
Local response (comparing FE and MT methods): (**a**) ${\dot{E}}_{xy}=10^{-2}s^{-1}$, (**b**) ${\dot{E}}_{xy}=10^{-3}s^{-1}$.

**Figure 13 polymers-14-03301-f013:**
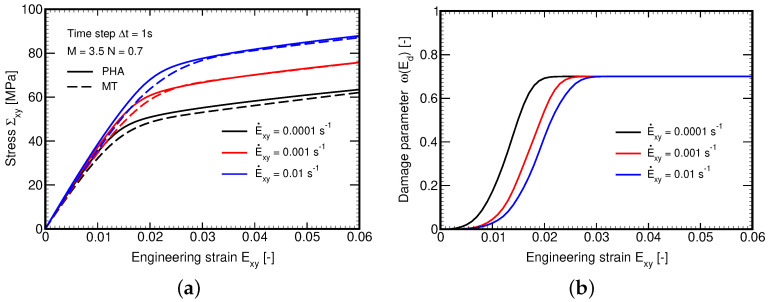
(**a**) Evolution of stress $\Sigma_{xy}$ for different strain rates with model parameters fitted to ${\dot{E}}_{xy}=0.001$ s${}^{-1}$: comparison of MT and PHA predictions, (**b**) Evolution of damage parameter ω for different strain rates.

**Figure 14 polymers-14-03301-f014:**
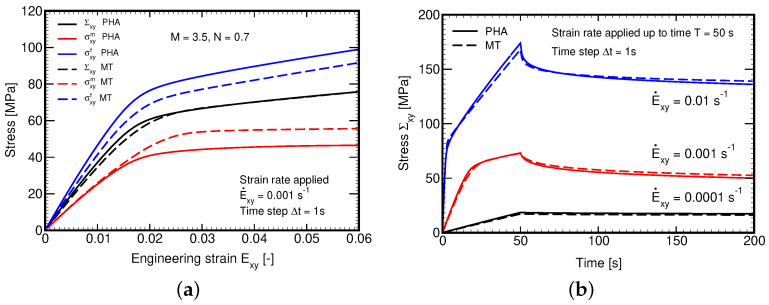
(**a**) Comparison of MT and PHA predictions for ${\dot{E}}_{xy}=$1 × 10^−3^ s${}^{-1}$: evolution of stresses within individual phases, (**b**) Relaxation test: time evolution of macroscopic strain $E_{xy}$.

**Figure 15 polymers-14-03301-f015:**
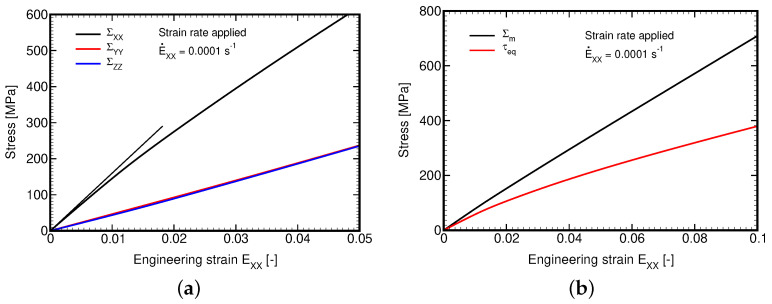
Plain weave composite loaded in tension by prescribed macroscopic strain rate ${\dot{E}}_{XX}=10^{-4}$s${}^{-1}$: (**a**) macroscopic stress strain curves, (**b**) evolution of macroscopic mean stress $\Sigma_{m}$ and equivalent deviatoric stress $\tau_{eq}$.

**Figure 16 polymers-14-03301-f016:**
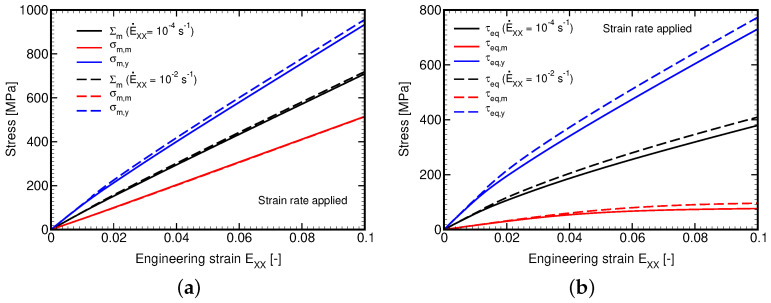
Plain weave composite loaded in tension by prescribed macroscopic strain rates ${\dot{E}}_{XX}=10^{-2}$s${}^{-1}$ and ${\dot{E}}_{XX}=10^{-4}$s${}^{-1}$: (**a**) mean stress, (**b**) equivalent deviatoric stress.

**Figure 17 polymers-14-03301-f017:**
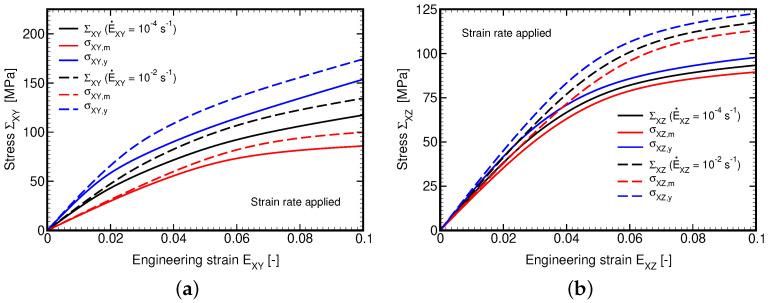
Plain weave composite loaded by prescribed macroscopic in-plane and out-of-plane shear strain rate ${\dot{E}}_{Xi}=10^{-2}$s${}^{-1}$ and ${\dot{E}}_{Xi}=10^{-4}$s${}^{-1}$: (**a**) $\Sigma_{XY}\times E_{XY}$, (**b**) $\Sigma_{XZ}\times E_{XZ}$.

**Table 1 polymers-14-03301-t001:** Chemical specification of epoxy resin and hardeners (25 °C), see [11,25].

Parameter	L285 Resin	H508 Hardener	H500 Hardener
Viscosity [Pa·s]	0.6–0.9	0.007–0.011	0.2–0.35
Density [g/cm${}^{3}$]	1.18–1.23	0.93–0.96	1.0–1.06
Molecular weight [g]	182–192	-	-
Amine value [mgKOH/g]	-	450–500	350–400
Average EP-Value	0.63	-	-

**Table 2 polymers-14-03301-t002:** Geometry of specimens.

Parameter (H508)	Width [mm]	Thickness [mm]	Cross-section area [mm${}^{2}$]
Mean	5.04	2.76	13.90
Standard deviation	0.02	0.05	0.28
Parameter (H500)	Width [mm]	Thickness [mm]	Cross-section area [mm${}^{2}$]
Mean	4.11	2.95	12.13
Standard deviation	0.28	0.14	0.98

**Table 3 polymers-14-03301-t003:** Indentation results of basalt fibers.

Direction	Reduced Mod. $E_{r}$ [GPa]	Young’s Mod. *E* [GPa]	Hardness *H* [GPa]	Indentation Depth $h_{c}$ [nm]
Longitudinal	69.94 ± 7.80	69.68 ± 7.40	5.58 ± 0.68	160.51 ± 10.00
Transverse	65.18 ± 7.32	64.82 ± 7.03	5.29 ± 1.19	114.52 ± 14.15

**Table 4 polymers-14-03301-t004:** Phase elastic properties. Elastic moduli are in [GPa].

Material	$G^{\mathrm{el}}$	$K^{\mathrm{el}}$	$E_{L}$	$E_{T}$	$G_{L}$	$\nu_{L}$	$\nu_{T}$
**Matrix**	1.42	3.96	-	-	-	-	-
**Fibers**	-	-	69.7	64.8	28.1	0.40	0.24

**Table 5 polymers-14-03301-t005:** Parameters of Maxwell chain model—L285 epoxy resin.

μ	$\tau_{\mu }$ [s]	$J_{\mu }$ [MPa${}^{-1}$]	$\theta_{\mu }$ [MPa·s]	$G_{\mu }$ [MPa]
1	$1\times 10^{-5}$	7.047033 × $10^{-4}$	9.655035 × $10^{-2}$	4.915541 × $10^{1}$
2	0.1	2.507001 × $10^{-5}$	9.672027 × $10^{-1}$	4.501054 × $10^{1}$
3	1	2.469968 × $10^{-5}$	9.569420 × $10^{+0}$	5.732147 × $10^{1}$
4	10	3.377764 × $10^{-5}$	9.321383 × $10^{+1}$	8.651850 × $10^{1}$
5	100	5.703039 × $10^{-5}$	9.312438 × $10^{+2}$	8.338651 × $10^{1}$
6	1000	6.067732 × $10^{-5}$	9.073146 × $10^{+3}$	1.324220 × $10^{2}$
7	10,000	6.906896 × $10^{-5}$	3.361408 × $10^{+4}$	7.832536 × $10^{2}$
8	100,000	8.026744 × $10^{-4}$	1.565791 × $10^{+5}$	1.294026 × $10^{2}$
9	1,000,000	1.724650 × $10^{-2}$	7.293674 × $10^{+5}$	5.256629 × $10^{1}$

**Table 6 polymers-14-03301-t006:** Textile ply effective properties from FEM homogenization. Elastic moduli are in [GPa] ($\nu_{XY}=\nu_{YX},\nu_{XZ}=\nu_{YZ},\nu_{ZX}=\nu_{ZY}$).

$E_{\mathrm{XX}}$	$E_{\mathrm{YY}}$	$E_{\mathrm{ZZ}}$	$G_{\mathrm{YZ}}$	$G_{\mathrm{XZ}}$	$G_{\mathrm{XY}}$	$\nu_{\mathrm{XY}}$	$\nu_{\mathrm{XZ}}$	$\nu_{\mathrm{ZY}}$
12.4	12.4	6.6	2.1	2.1	2.6	0.19	0.41	0.22

## Data Availability

Not applicable.

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
