# Peer review of "Computational Modeling of Polymer Matrix Based Textile Composites"

_polymers, 2022, doi:10.3390/polym14163301_

Round 1

Reviewer 1 Report

The computational modeling of polymer matrix is a very fundamental and important research area. In this manuscript, author use several model to simulate the textile reinforced epoxy resin composite. They remedied the accurate of FEM model and proved the simulation could replace the actual laboratory measurements in a field of plastic matrix. Then it was confirmed that the multi-scale textile-reinforced plastic composites could be predicted for a macroscopic response of tension and shearing measurements. 

The manuscript is well organized and written. The improved FEM model can be useful in application of virtual measurement of plastic composites. I suggest this work can be published in the journal as current state.

Author Response

the authors would like to thank the reviewers for their valuable comments to improve the paper.

The attached .pdf file highlights the changes made.

Reviewer 2 Report

A very interesting topic of multiscale analysis of multiscale analysis of a plain weave reinforced composite made of basalt fabrics bonded to a epoxy resin was presented. The chosen fabric is made of basalt fibres, which have an increasing application and are therefore very topical. The modelling methodology was presented using various numerical simulations. This is valuable data for all those involved in the development of new materials. This makes the paper very interesting. Although it is a purely computational study that should be validated experimentally, the paper is original and very valuable.

The paper is conceptually well done and no changes are needed.

The recommendation to the editorial board is to publish the paper.

Author Response

(The authors gave the same response as above.)

Reviewer 3 Report

Dear Authors,

Please find the enclosed file.

Author Response

(The authors gave the same response as above.)
